# Serotonergic neurons signal reward and punishment on multiple timescales

**Jeremiah Y Cohen[1,2]\*, Mackenzie W Amoroso[1], Naoshige Uchida[1]**

[1]Department of Molecular and Cellular Biology, Center for Brain Science, Harvard University, Cambridge, United States; [2]Solomon H. Snyder Department of Neuroscience, Brain Science Institute, The Johns Hopkins University School of Medicine, Baltimore, United States

**Abstract** Serotonin's function in the brain is unclear. One challenge in testing the numerous hypotheses about serotonin's function has been observing the activity of identified serotonergic neurons in animals engaged in behavioral tasks. We recorded the activity of dorsal raphe neurons while mice experienced a task in which rewards and punishments varied across blocks of trials. We 'tagged' serotonergic neurons with the light-sensitive protein channelrhodopsin-2 and identified them based on their responses to light. We found three main features of serotonergic neuron activity: (1) a large fraction of serotonergic neurons modulated their tonic firing rates over the course of minutes during reward vs punishment blocks; (2) most were phasically excited by punishments; and (3) a subset was phasically excited by reward-predicting cues. By contrast, dopaminergic neurons did not show firing rate changes across blocks of trials. These results suggest that serotonergic neurons signal information about reward and punishment on multiple timescales.

## Introduction

Reward and punishment play critical roles in shaping animal behavior over short and long timescales (*Rolls, 2005*; *Somerville et al., 2013*). On short timescales, moment-to-moment expectations of reward or punishment increase or decrease motivation to perform a specific action associated with the reward or punishment. On long timescales, repeated exposure to reward or punishment can elicit long-lasting positive or negative emotional states (often called 'mood'), that can increase or decrease the frequency of performing reward-seeking actions more generally (*Niv et al., 2006*; *Cools et al., 2011*; *Somerville et al., 2013*; *Wang et al., 2013*).

The midbrain raphe nuclei contain the majority of forebrain-projecting serotonergic neurons in mammals (*Jacobs and Azmitia, 1992*). This small population of neurons (approximately 9000 in the mouse dorsal raphe; *Ishimura et al., 1988*) projects to almost the entire forebrain (*Azmitia and Segal, 1978*; *Moore et al., 1978*; *Steinbusch, 1981*; *O'Hearn and Molliver, 1984*; *Vertes, 1991*; *Gagnon and Parent, 2014*). The diffuse projection targets of the serotonergic system have led to many theories about its function.

Serotonin has been proposed to be involved in processing reward and punishment (*Maswood et al., 1998*; *Daw et al., 2002*; *Maier and Watkins, 2005*; *Nakamura et al., 2008*; *Ranade and Mainen, 2009*; *Tops et al., 2009*; *Cools et al., 2011*; *Amo et al., 2014*). One theory proposes that serotonin regulates aversive learning and negative motivation in response to punishments (*Soubrié, 1986*; *Deakin and Graeff, 1991*; *Daw et al., 2002*; *Dayan and Huys, 2009*). According to this theory, serotonin opposes the positive reinforcement and behavioral activation regulated by dopamine. Whereas dopaminergic neurons signal appetitive prediction errors (*Schultz et al., 1997*), serotonergic neurons could signal punishments, thereby adjusting future behavior to avoid those punishments or inhibiting specific actions that are associated with punishments. This theory has support from lesion,

\*For correspondence: jeremiah.cohen@jhmi.edu

**Competing interests:** The authors declare that no competing interests exist.

**eLife digest** Rewards and punishments can both encourage animals to change their immediate behavior and influence their mood over a longer term, particularly when given repeatedly. A region of the brain that increases its activity in response to rewards and punishments also contains many neurons that communicate with each other by releasing a chemical called serotonin. This chemical is commonly thought to produce feelings of happiness; however, it remains unclear exactly how these particular 'serotonergic' neurons help to process rewards and punishments.

The ideal way to work out the role that a type of neuron plays in a behavior is to measure its electrical activity as the behavior is being performed. However, it is difficult to distinguish the activity of serotonergic neurons from the activity of the non-serotonergic neurons around them. To overcome this problem, Cohen et al. used viruses to force serotonergic neurons to make a type of ion channel that produces electrical currents in response to light. Shining light on these neurons via optical fibers and then measuring the neurons' responses helped to develop criteria that can identify which responses are generated by the serotonergic neurons.

Cohen et al. then recorded the activity of serotonergic neurons in thirsty mice as they experienced a series of rewards (for example, a drop of water) or punishments (such as a puff of air to the eye). Each reward or punishment was preceded by a distinct odor, so that the mice learned to anticipate what was coming.

These experiments revealed that serotonergic neurons respond to rewards and punishments by changing two aspects of their electrical activity: by producing short bursts of high activity, and by altering their baseline activity. Some of the serotonergic neurons fired rapidly in response to punishments, but not rewards; others fired rapidly when the mice detected a scent that meant that a reward was about to be given. The average level of reward or punishment the mice received also affected the baseline activity of many of the serotonergic neurons; this effect lasted for several minutes.

Overall, Cohen et al. suggest that serotonergic neurons can affect how mice respond to rewards or punishments in both the short and long term. Future experiments should aim to understand the diversity of the signals that Cohen et al. observed, and to determine how these signals are used to drive behavior. Ultimately, understanding how neural circuits made up of different types of cells work may aid in understanding the neural basis of behavior.

stimulation, tryptophan depletion, and pharmacological studies (*Tye et al., 1977*; *Graeff and Silveira Filho, 1978*; *Liu and Ikemoto, 2007*; *Crockett et al., 2009*; *Shin and Ikemoto, 2010*), but there is little evidence that serotonergic neurons signal punishments in awake animals (cf. *Aghajanian et al., 1978*; *Montagne-Clavel et al., 1995*; *Schweimer and Ungless, 2010*).

A second theory proposes that serotonin signals global reward states, such as tracking average reward (*Daw et al., 2002*) and modulating mood (*Savitz et al., 2009*). Here, serotonin is thought to provide its diffuse targets with long-term signals about the value of the environment, which, at the extreme, is correlated with changes in mood. These long-term signals inhibit behavior or increase the vigor of taking actions in a relatively action-general manner. This theory has support from clinical observations (*Fava and Kendler, 2000*) and genetic studies (*Donaldson et al., 2013*). However, there is little neurophysiological evidence for such a function.

A third theory proposes that serotonin is involved in waiting for reward (*Miyazaki et al., 2011a*, *2011b*, *2014*; *Fonseca et al., 2015*). In this theory, activation of serotonergic neurons promotes patience (*Miyazaki et al., 2014*) or slows movements that allow an animal to wait for a delayed reward. This theory could explain how waiting for reward could be linked to behavioral inhibition (*Soubrié, 1986*; *Fonseca et al., 2015*).

Clarifying whether and how serotonin exerts these functions requires understanding how serotonergic neuron firing correlates with both global features of the environment, such as changes in average reward value, and with punishments. These data have been challenging to collect because of the heterogeneity of neurons in and around the raphe nuclei. About two-thirds of dorsal raphe neurons are serotonergic, but others contain GABA, glutamate, dopamine, acetylcholine, or various peptides (*Hökfelt et al., 2000*; *Commons, 2009*; *Fu et al., 2010*; *Hioki et al., 2010*). Many previous

studies have relied on spike waveform characteristics to identify serotonergic neurons in extracellular recordings. This approach may lead to false positives and misses, however (*Allers and Sharp, 2003*; *Kirby et al., 2003*; *Marinelli et al., 2004*; *Urbain et al., 2006*; *Hajós et al., 2007*; *Schweimer et al., 2011*).

Given this chemical diversity and the difficulty of identifying neuron types in extracellular recordings, it would be unsurprising to find corresponding physiological diversity. Indeed, previous studies found significant heterogeneity in the activity of dorsal raphe neurons in relation to behavioral tasks (*Fornal et al., 1996*; *Nakamura et al., 2008*; *Ranade and Mainen, 2009*; *Bromberg-Martin et al., 2010*; *Miyazaki et al., 2011a*; *Inaba et al., 2013*; *Li et al., 2013*). One set of experiments demonstrated that dorsal raphe neurons fired in a reward-value-dependent manner during trials of a saccade-to-target task, but that the modulation disappeared before the next trial (*Nakamura et al., 2008*; *Bromberg-Martin et al., 2010*). Another set of experiments showed tonic firing rate modulations correlating with waiting for reward from putative dorsal raphe serotonergic neurons (*Miyazaki et al., 2011a*). These studies suggested that serotonergic neurons signal reward expectation or waiting time (or patience) on the scale of hundreds of milliseconds, but does not determine whether serotonin could be involved in longer-term changes in value.

In the present study, we sought to test whether dorsal raphe serotonergic neuron firing correlated with rewards and punishments on different timescales. To address this question, we used a task in which reward value changed on slow and fast timescales. Importantly, we recorded from optogenetically-identified serotonergic neurons. Our results show that serotonergic neurons signal information about reward and punishment across multiple timescales.

## Results

### Behavioral task

We classically conditioned head-fixed, thirsty mice with different odor cues that predicted a reward (water), neutral outcome (nothing), or punishment (a puff of air delivered to the animal's face; *Cohen et al., 2012*). Each behavioral trial began with a conditioned stimulus (CS; an odor, 1 s), followed by a 1-s delay and an unconditioned stimulus (US; the outcome; *Figure 1A*). Mice licked toward the water-delivery tube in the delay period before rewards arrived, but not during neutral or punishment trials, indicating that they had learned the CS-US associations (analysis of variance (ANOVA), t-tests, p < 0.001 for each session; *Figure 1B*). We varied the structure of the task so that animals received blocks of 10 reward trials, alternating with 10 punishment trials, with a tone indicating transitions between blocks (or, in 35% of sessions, block order was random; *Figure 1C*; block duration mean ± S.D., 1.36 ± 0.30 min). Mice licked significantly more during the tone preceding reward vs punishment blocks. Because the tone was the same preceding both block types, this indicates that they attended to the block structure (Wilcoxon signed-rank test, p < 0.01; *Figure 1—figure supplement 1*). In each block of some sessions, the reward or punishment trial was replaced by a neutral trial with probability 0.1. To ensure that the time of onset of each trial could not be predicted following the end of the previous one, we used an exponentially-distributed (flat hazard rate) inter-trial interval (ITI). The ITIs lasted, on average, longer than twice the duration of trials (ITI, 6.41 ± 1.88 s, mean ± S.D.). Using this task, we could study neuronal responses on fast (CS and US) and slow (across blocks) timescales.

To determine whether mice treated air puffs as punishments, we performed a behavioral experiment in which mice were given free choices between a water reward and a water reward delivered simultaneously with an air puff. Mice reliably chose the water reward without the air puff, confirming that air puffs act as a negative reinforcement (*Figure 1—figure supplement 2*).

### Identifying serotonergic neurons

We recorded the activity of 149 dorsal raphe neurons while mice performed the conditioning task (6 mice, 24.8 ± 6.1 neurons per mouse, mean ± S.E.M.). We expressed channelrhodopsin-2 (ChR2), a light-gated cation channel, in serotonergic neurons by injecting an adeno-associated virus containing FLEX-ChR2 (AAV5-EF1α-DIO-hChR2(H134R)-EYFP-WPRE-pA) into the dorsal raphe of transgenic mice expressing Cre recombinase under the control of the promoter of the serotonin transporter gene (*Slc6a4*; *Sert*-Cre mice). Expression was specific (of 260 ChR2-EYFP-positive cells, 6 cells, or 2.3%, were 5-HT-negative) and efficient (of 314 5-HT-positive cells, 254, or 80.9%, were ChR2-EYFP-positive; *Figure 1D*).

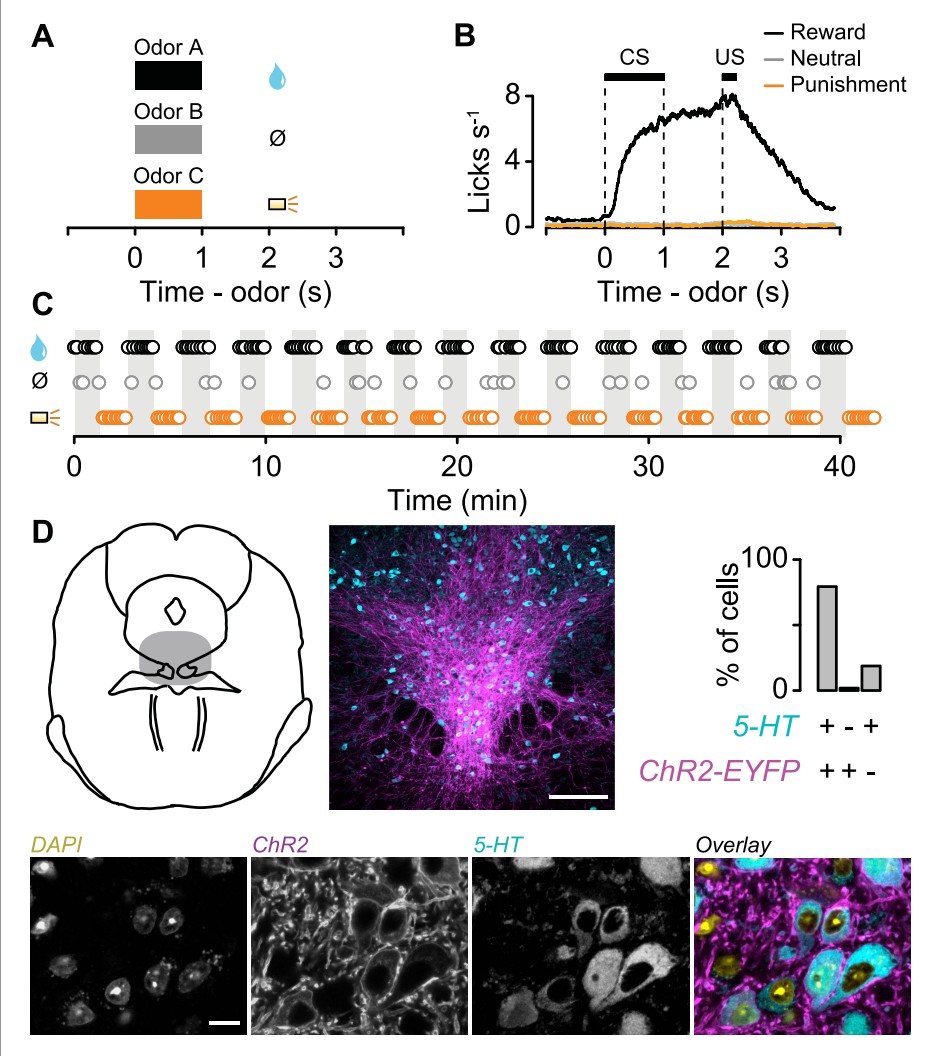

**Figure 1**. Behavioral task. (**A**) Structure of individual trials. (**B**) Average lick rate for all animals during each trial type. (**C**) Representative sequence of trials from one experiment. Each point represents an odor cue. Shaded regions indicate reward blocks. (**D**) Schematic of midbrain indicating recording sites (shaded), with low and high magnification of 5-HT labeling (cyan), ChR2-EYFP (magenta), nuclei (DAPI), and their overlay. Scale bars are 100 μm and 10 μm for low and high magnification, respectively.

The following figure supplements are available for figure 1:

**Figure supplement 1**. Histogram of lick rate during tones indicating block transitions, for experiments in which reward blocks alternated with punishment blocks.

**Figure supplement 2**. Mice treat air puffs as punishments.

For each neuron, we measured the response to light stimulation and the shape of spontaneous spikes (*Figure 2A,B*). We reasoned that for a neuron to be identified as responding to light stimulation, it must first show a significant response to the stimulus (quantified here as light-evoked energy, defined as the integral of the squared voltage values $\int v^2 dt$). Second, to ensure that the neuron under observation, rather than a population of nearby ChR2-expressing serotonergic neurons, responded to light stimulation, we verified that the response to light stimulation matched the shape of spontaneous spikes. We calculated the distance between the spontaneous spike waveform and light-evoked voltage response and plotted it against the energy of light-evoked response for each

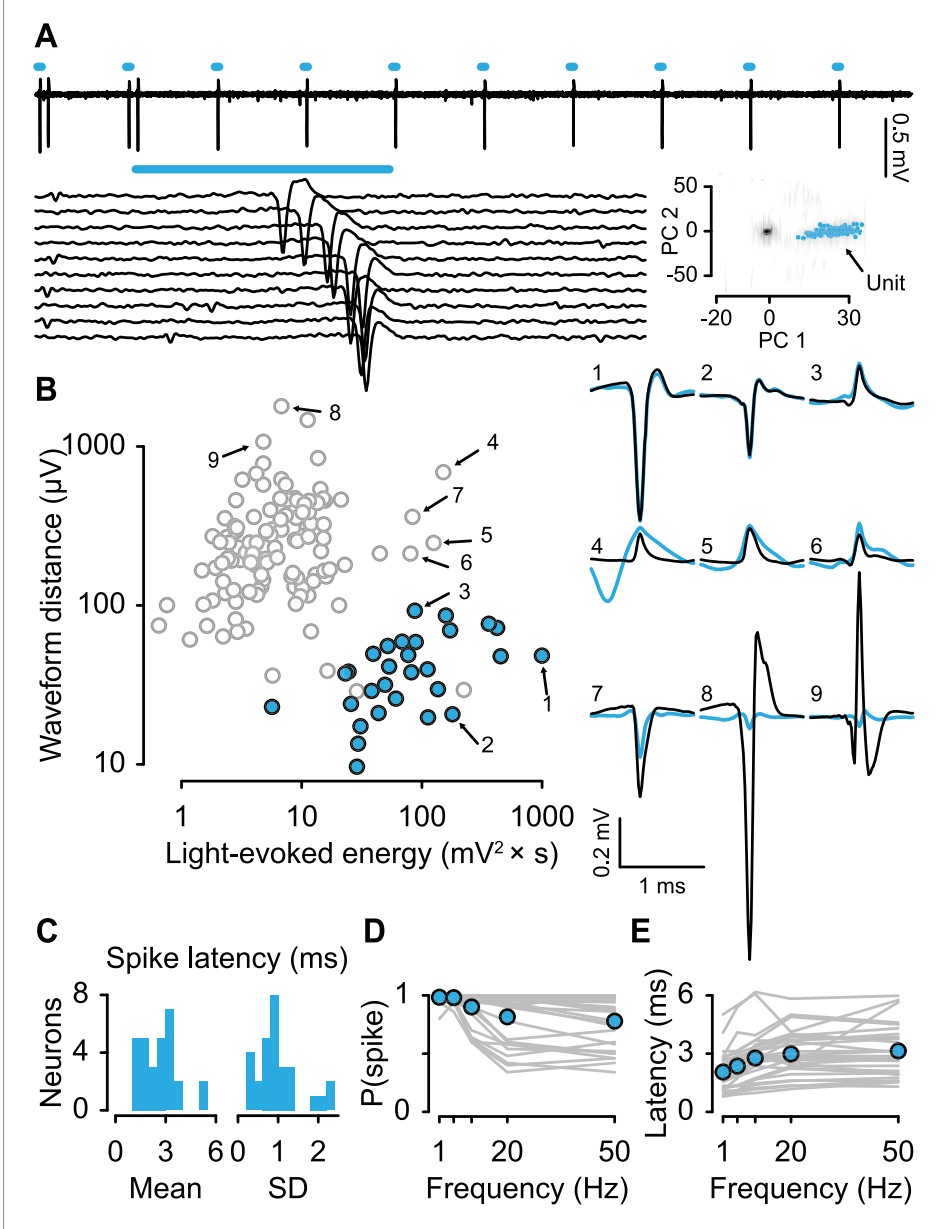

**Figure 2.** Identifying serotonergic neurons. (**A**) Example voltage trace from 10 pulses of 10-Hz light stimulation (cyan bars; light duration, 5 ms). Each light-triggered spike is shown below. The lower right is the first two principal components of all waveforms from one tetrode wire, showing the neuron's ('unit') isolation quality, with 100 randomly-chosen light-evoked spikes in cyan. (**B**) Quantification of light-evoked responses to identify serotonergic neurons (filled points). Abscissa: energy (integral of the squared voltage values) of the light-evoked response from each neuron. Ordinate: Euclidean distance between the mean spontaneous spike and the light-evoked response. Example neurons are shown to the right (black, spontaneous spikes; cyan, light-evoked voltages; SD of spike waveforms are smaller than line thicknesses). Note that three unfilled points in the lower-right cluster are not considered identified serotonergic neurons because of low probability of firing in response to light stimulation. (**C**) Mean and SD of the light-evoked spike latency for identified serotonergic neurons. Probability (**D**) and latency (**E**) of light-evoked spikes from serotonergic neurons as a function of stimulation frequency (filled points are means across neurons).

recording (**Figure 2B**). Using an expectation-maximization clustering method, we observed two distinct clusters: one that showed significant responses to light pulses and one that did not. 29 neurons fell into the former cluster (filled cyan points in **Figure 2B**). Three points in that cluster were

not considered identified serotonergic neurons because they did not reliably respond to light stimulation. Consistent with direct light activation rather than indirect synaptic activation, all 29 neurons showed fast light-evoked spikes (*Figure 2C*) and followed high-frequency stimulation (*Figure 2D,E*). These properties indicate that these 29 neurons expressed ChR2 (henceforth called 'serotonergic neurons'; 5 mice, 5.8 ± 1.5 neurons per mouse, mean ± S.E.M.).

## All serotonergic neurons show task-related activity

The structure of the behavioral task allowed us to ask whether serotonergic neuron firing correlated with short-term (rewards, punishments, and the cues that predicted them within a trial) or long-term (blocks of reward vs punishment trials) changes in the environment.

We first asked whether serotonergic neuron firing rates were significantly modulated during the behavioral task. We performed an ANOVA on the trial-by-trial firing rates during the baseline period (1 s before odor onset), CS period (from odor onset to odor offset), delay (from odor offset to outcome onset), and US period (from outcome onset to 500 ms after outcome onset). The factors were task epoch (baseline, CS, delay, or US) and outcome type. All 29 serotonergic neurons exhibited task-related modulations in firing rate (ANOVA, all $p < 0.01$).

## Tonic firing modulation by long-term values

Next, we examined the responses of serotonergic neurons in detail. We observed three main features of the firing patterns of serotonergic neurons. First, a large fraction (41%) of serotonergic neurons fired at a higher or lower rate during the ITIs of reward blocks vs punishment blocks (*Figure 3A,B*; *Figure 3—figure supplement 1*). That is, even before a particular trial began, these neurons fired at a rate that correlated with the value (reward vs punishment) of the block. Remarkably, this response persisted across minutes. To quantify this observation, we calculated the firing rate in the 2 s before the start of each trial during reward and punishment blocks. 12 of 29 serotonergic neurons showed significantly different pre-trial firing rates between reward and punishment blocks: 7 were more excited during reward blocks, 5 were more excited during punishment blocks (*Figure 3C,D*; Wilcoxon rank sum tests, $p < 0.05$). Interestingly, this tonic signal appeared to build up or down slowly within blocks, rather than sharply increasing or decreasing in response to block transitions (*Figure 3—figure supplement 2*). This tonic signal did not depend on the duration of ITIs (Wilcoxon rank sum tests, all $p > 0.3$). In addition, 16 of 29 serotonergic neurons displayed gradually decreasing firing rates over the course of the experiment (*Figure 3—figure supplement 3*).

To compare this effect to the response of dopaminergic neurons—which have been proposed to be involved in long-term value-related signaling (*Niv et al., 2006*; *Cools et al., 2011*)—in the same task, we recorded the activity of 28 ventral tegmental area (VTA) neurons, 15 of which were identified as dopaminergic (by ChR2 tagging, as described above), and 13 of which were putatively dopaminergic based on their task-related responses (4 mice, 7.0 ± 2.0 neurons per mouse, mean ± S.E.M.). None of the putative or identified dopaminergic neurons showed significantly different pre-trial firing rates between reward and punishment blocks (Wilcoxon rank sum tests, all $p > 0.5$; *Figure 3E–H*). Most dopaminergic neurons were excited by predicted rewards to varying degrees (see *Figure 3—figure supplement 4* for a more 'canonical' example), similar to prior observations in classical conditioning tasks with trace delays in mice (*Cohen et al., 2012*) and monkeys (*Fiorillo et al., 2008*).

## Serotonergic neurons are phasically excited or inhibited by reward-predicting cues or punishments

The second main feature of serotonergic neuron activity we observed was their response to punishments (*Figure 4A–C*, *Figure 4—figure supplement 1*). Previous studies found identified serotonergic neurons to be excited or inhibited by punishments in anesthetized animals (*Aghajanian et al., 1978*; *Montagne-Clavel et al., 1995*; *Schweimer and Ungless, 2010*). To test whether punishments modulated serotonergic firing, we calculated the area under the receiver operating characteristic (auROC) curve in sliding 100-ms windows for each neuron, comparing each window during the trial to the baseline firing rate (1 s before punishment trials; *Figure 4B*). The auROC quantifies the discriminability of the two firing rate distributions. Values of 0.5 (black) indicate no change in firing rate relative to baseline. Values greater than 0.5 (yellow) indicate increases in firing

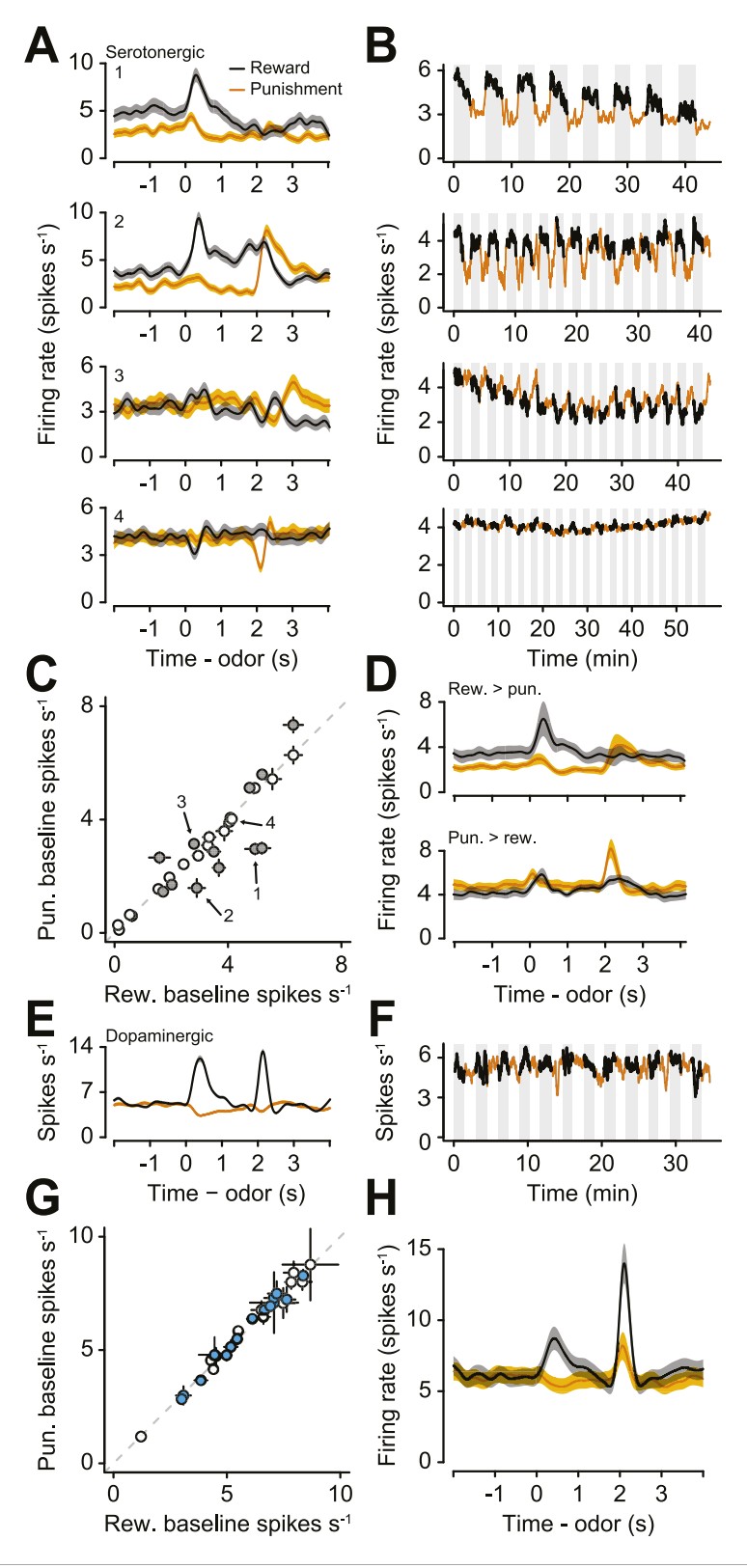

**Figure 3**. A population of serotonergic neurons is more or less active during blocks of reward trials than punishment trials. (**A**) Average firing rates of four example serotonergic neurons during reward trials (black) and punishment trials (orange). Shaded regions denote S.E.M. Note the higher pre-trial firing rate during reward trials than punishment

*Figure 3. continued on next page*

*Figure 3. Continued*

trials in the top two neurons and the higher firing rate during punishment vs reward blocks in the third neuron. (**B**) Firing rate of the same four neurons across the timecourse of the experiment. Note the slow (across minutes) fluctuations in firing rate in the top three neurons that correlated with block type (reward: black, shaded regions; punishment: orange). (**C**) Mean ±95% confidence intervals around firing rates during the baseline epoch for punishment vs reward blocks for each serotonergic neuron (significant data points are filled). Examples from (**A**) are labeled. (**D**) Average firing rates of serotonergic neurons with significantly higher baseline firing rates during reward (top) or punishment (bottom) blocks. (**E**) Average firing rate of an example identified dopaminergic neuron during reward and punishment trials. (**F**) Firing rate across the timecourse of the experiment for the dopaminergic neuron in (**E**). (**G**) Mean ±95% confidence intervals around firing rates during the baseline epoch for punishment vs reward blocks for each dopaminergic neuron (identified in cyan, putative in white). (**H**) Average firing rate of dopaminergic (identified and putative) neurons during reward and punishment trials.

The following figure supplements are available for figure 3:

**Figure supplement 1**. Raster plots showing spike times during 40 trials for the four example neurons in *Figure 3A*.

**Figure supplement 2**. (**A**) Normalized mean ± S.E.M. firing rate within reward and punishment blocks for the positive-coding serotonergic neurons.

**Figure supplement 3**. (**A**) Trial-by-trial scatter plot of lick rate against spike rate for an example serotonergic neuron.

**Figure supplement 4**. Example activity of a dopaminergic neuron with a smaller response to predicted reward than unpredicted reward-predicting cue.

rate relative to baseline, while values less than 0.5 (blue) indicate decreases in firing rate relative to baseline. Most serotonergic neurons (28 out of 29) responded phasically to punishments: 22 were excited, 6 were inhibited (*Figure 4A–C*; Wilcoxon rank sum tests in the 500 ms after punishment onset, $p < 0.05$). This response was transient, lasting less than 500 ms for most neurons ($315 \pm 140$ ms, mean $\pm$ S.D.).

The third main feature of serotonergic neuron activity we observed was their response to reward-predicting cues (*Figure 4D–F*). We calculated the auROC, comparing the baseline (1 s before reward trials) to each 100-ms window during the trial. About half of serotonergic neurons (15 out of 29) were phasically excited by reward-predicting cues (*Figure 4D–F*; Wilcoxon rank sum tests during the 1 s of odor presentation, $p < 0.05$). Very few neurons (2 out of 29; $p < 0.05$) were inhibited by reward-predicting cues. The duration of excitation was brief, lasting less than 500 ms ($235 \pm 194$ ms, mean $\pm$ S.D.). For 9 of these 15 neurons, the response to a reward-predicting CS was significantly greater than the response to a punishment-predicting CS (Wilcoxon rank sum tests, $p < 0.05$). The time of peak response during a reward-predicting CS was significantly shorter for serotonergic than dopaminergic neurons (mean $\pm$ S.E.M., $331 \pm 15.7$ ms for serotonergic neurons, $388 \pm 12.7$ ms for dopaminergic neurons, Wilcoxon rank sum test, $p < 0.05$).

Recent work has provided conflicting views on the role of the dorsal raphe in reward behavior (*Liu and Ikemoto, 2007*; *Shin and Ikemoto, 2010*; *Liu et al., 2014*; *McDevitt et al., 2014*; *Miyazaki et al., 2014*; *Qi et al., 2014*; *Fonseca et al., 2015*). To compare the phasic reward-related responses during the task to responses to unexpected rewards, we examined dopaminergic and serotonergic responses to unexpected reward (delivered at random times prior to the task in 5 *Slc6a4*-Cre mice and 3 *Slc6a3*-Cre mice). Whereas dopaminergic neurons showed a large, phasic excitation in response to unexpected rewards, a subset of serotonergic neurons was weakly and slowly, but significantly, excited (Wilcoxon rank sum tests, $p < 0.05$ for 9 of 10 dopaminergic neurons and 11 of 29 serotonergic neurons; *Figure 4G,H*). We compared neuronal responses to unexpected rewards to responses to expected rewards in the context of the task. Dopaminergic neurons showed larger responses to unexpected vs expected rewards (*Figure 4I*), similar to previous observations (e.g., *Schultz et al., 1997*; *Cohen et al., 2012*). Interestingly, serotonergic neurons showed a weak but significantly larger response to unexpected vs expected rewards (*Figure 4I*).

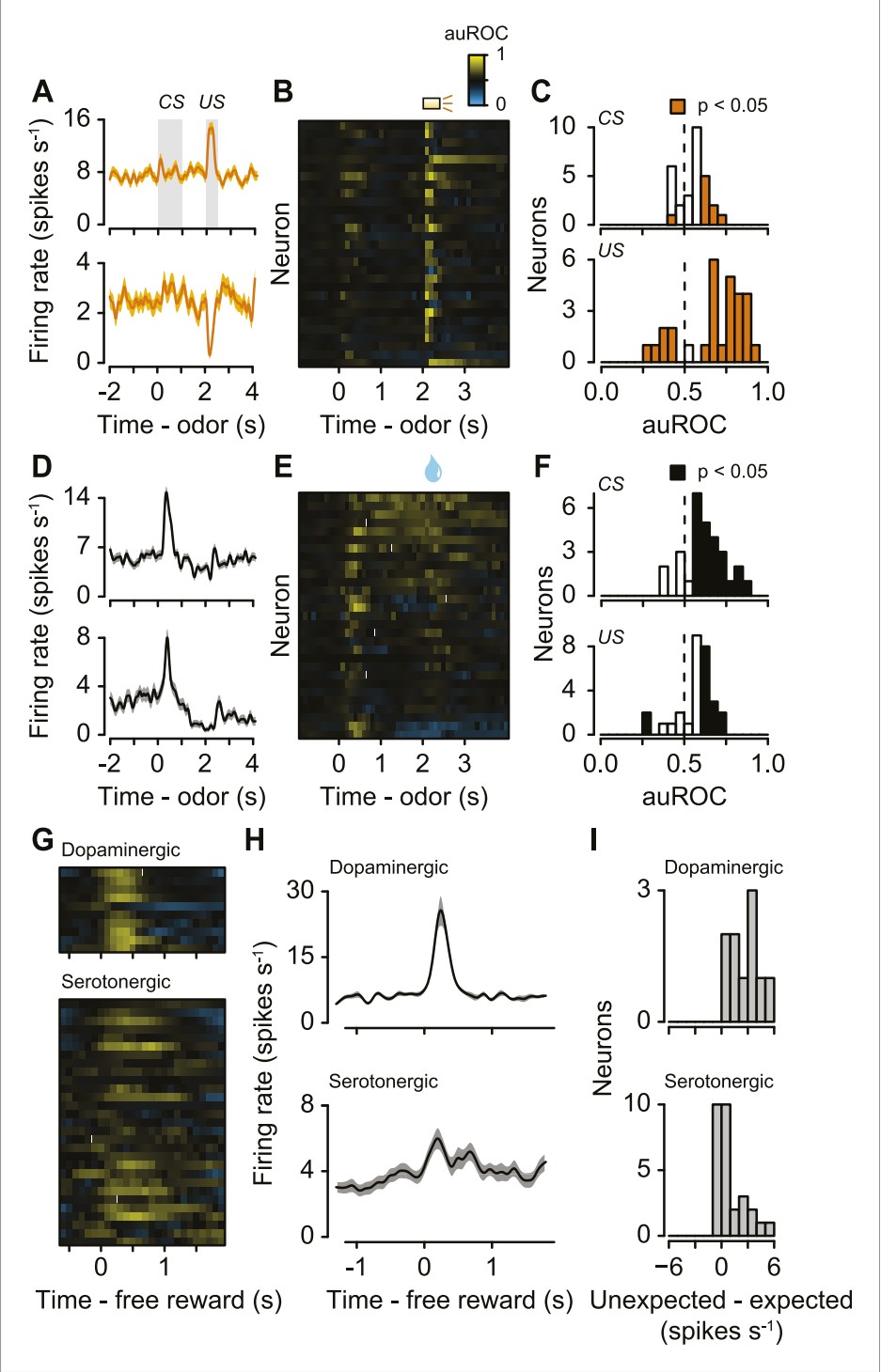

**Figure 4**. Serotonergic neurons are briefly excited or inhibited by punishments or reward-predicting cues. (**A**) Average firing rates of two example serotonergic neurons during punishment trials. CS and US analysis windows are shaded in gray. (**B**) Area under the ROC curve for punishment trials for all serotonergic neurons, sorted by the sum of the auROC for reward trials in (**E**). Yellow indicates excitation, blue indicates inhibition, and black indicates no change relative to baseline. (**C**) Histogram of changes in firing rate relative to baseline during the CS and US epochs of punishment trials. (**D**) Average firing rates of two example serotonergic neurons during reward trials. (**E**) Area under the ROC curve for reward trials for all serotonergic neurons, sorted by their sum. (**F**) Histogram of changes in firing rate relative to baseline during the CS and US epochs of reward trials. (**G**) Area under the ROC curve for free reward for dopaminergic and serotonergic neurons. (**H**) Average firing rate of dopaminergic and serotonergic
*Figure 4. continued on next page*

*Figure 4. Continued*

neurons around free reward (shaded curves show S.E.M.). (**I**) Histograms of average differences between mean firing rates during expected vs unexpected rewards for serotonergic and dopaminergic neurons.
The following figure supplement is available for figure 4:

**Figure supplement 1**. (**A**) Area under the ROC curve for reward (water) vs punishment (air puff) trials for serotonergic neurons, sorted by the sum of the auROC.

## Correlations between serotonergic response features

Next, we tested whether these three features were correlated within the population of serotonergic neurons. We found that the difference in pre-trial firing rate during reward vs punishment blocks positively correlated with the difference in response to reward- vs punishment-predicting CS; neurons with higher excitation for reward than punishment CS tended to fire at a higher rate before reward than punishment trials (*Figure 5A*; Pearson's $r = 0.85$, $t_{27} = 8.56$, $p < 0.001$). This demonstrates that the same serotonergic neurons can multiplex different signals about reward and punishment on different timescales.

The difference in pre-trial firing rate (reward vs punishment) did not significantly correlate with the response to punishment (*Figure 5B*; Pearson's $r = 0.03$, $t_{27} = -0.17$, $p > 0.8$). The response to reward CS was significantly positively correlated with the response to punishment (*Figure 5C*; Pearson's $r = 0.60$, $t_{27} = 3.89$, $p < 0.01$).

## Serotonergic neuron background firing rates signal graded value

To test whether differences between firing rates during reward vs punishment blocks reflected value, as opposed to salience, we compared the firing rates during neutral trials to those during reward and punishment trials. Neurons with responses to neutral stimuli that fall in between those to rewards and punishments are defined as value-coding. Those with responses to rewards and punishments that are either both larger, or both smaller, than responses to neutral stimuli, are defined as salience-coding (*Matsumoto and Hikosaka, 2009*).

We asked whether serotonergic neurons were value-coding across blocks. For 16 serotonergic neurons, we modified the task to include blocks of neutral trials randomly interspersed among reward and punishment blocks (with equal probability). Eight of these displayed significantly different pre-trial

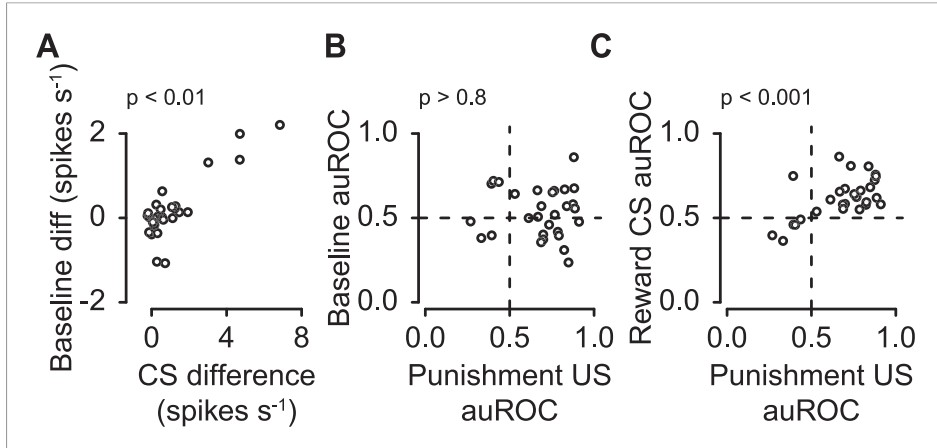

**Figure 5**. Correlations between serotonergic neuron response features. (**A**) Difference in firing rate during the 2-s pre-trial epoch vs the difference in firing rate during the CS (reward − punishment), corrected for baseline differences. (**B**) Area under the ROC curve for the 2-s pre-trial epoch during reward vs punishment trials plotted against area under the ROC curve during the punishment US epoch. (**C**) Area under the ROC curve for during the reward CS vs punishment US epochs.

firing rates during reward vs punishment blocks (*Figure 3C*). For 7 of these 8 neurons (3 of which displayed higher firing rates during reward blocks, 5 of which displayed higher firing rates during punishment blocks), pre-trial firing rates during neutral blocks did not fall outside of the bounds of those during reward and punishment blocks (*Figure 6A*; Tukey's Honest Significant Difference tests, $p > 0.05$). Only 1 of the 16 neurons had both significantly larger pre-trial firing rates during neutral blocks than during reward and punishment blocks. Thus, 15 of 16 serotonergic neurons were not salience-coding across blocks of trials.

So far, we have described differences in background firing rates as being value-related due to their differences between reward (water), neutral (no stimulus), and punishment (air puff) conditions. We asked whether responses across blocks were truly value-related, or, rather, due to differences in responses to the different sensory modalities for reward and punishment (gustatory vs

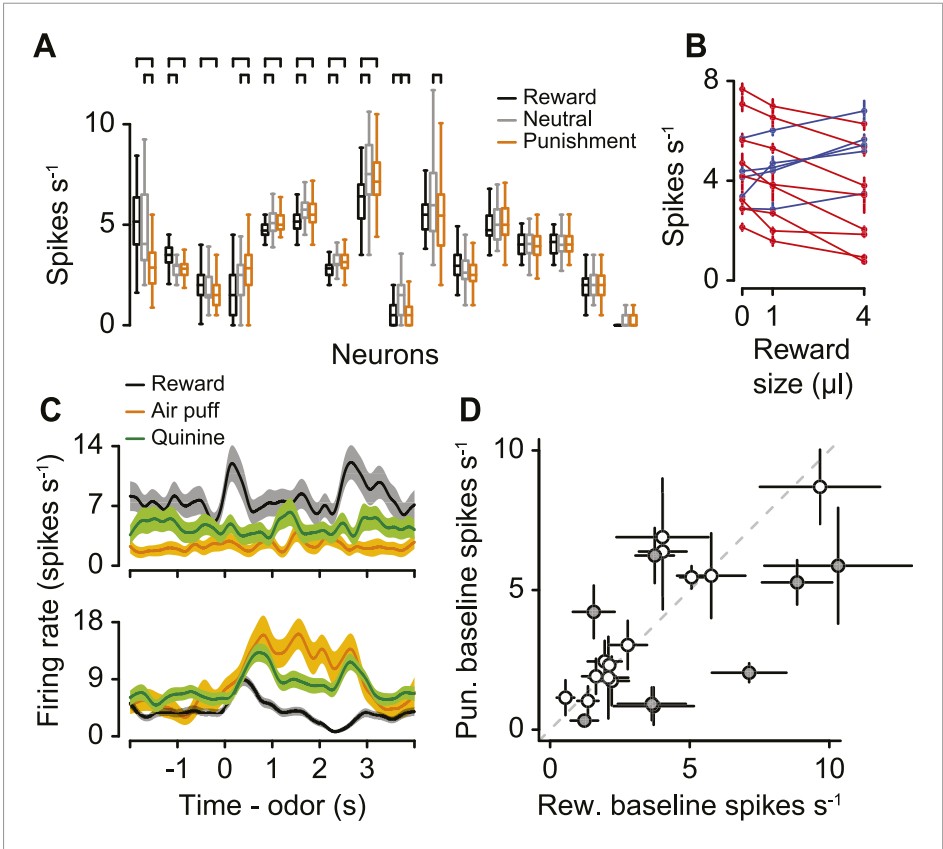

**Figure 6**. Serotonergic neuron background firing rates signal graded value. (**A**) Median (horizontal line), interquartile range (box), and 1.5 times interquartile range (whiskers) of pre-trial firing rates during reward (black), neutral (gray), and punishment (orange) blocks. Brackets indicate significant differences. (**B**) Mean ±95% confidence intervals pre-trial response during blocks of three reward sizes for 13 serotonergic neurons with strictly increasing (blue) or decreasing (red) firing rates as a function of reward size. (**C**) Mean ± S.E.M. firing rate of two example serotonergic neurons during reward (water or chocolate milk) and punishment (air puff, orange; quinine, green) trials. (**D**) Mean ± 95% confidence intervals around firing rates during the baseline epoch for punishment (quinine) vs reward (water or chocolate milk) blocks for each serotonergic neuron (significant data points are filled).

The following figure supplements are available for figure 6:

**Figure supplement 1**. Behavioral task.

**Figure supplement 2**. (**A**) Area under the ROC curve for reward, air-puff punishment, and quinine punishment trials for 21 serotonergic neurons.

somatosensory). We performed two experiments to address this. We recorded from 13 additional identified serotonergic neurons in which blocks contained trials with one of three reward sizes: zero, small (1 µl), or big (4 µl; *Figure 6—figure supplement 1*). If pre-trial, tonic responses are value-related, they should vary monotonically with reward size. Indeed, for each of the 13 neurons, firing rates were either strictly increasing (5 neurons) or strictly decreasing (8 neurons) with reward size (*Figure 6B*; Tukey's Honest Significant Difference tests, p > 0.05). Next, we recorded from 21 additional identified serotonergic neurons (4 mice, 5.3 ± 1.0 neurons per mouse, mean ± S.E.M.) in a similar task with four types of blocks: water or chocolate milk reward, neutral, air-puff punishment, and quinine punishment. Quinine, a bitter-tasting solution, is a punishment of the same sensory modality as a water reward. We found that 8 of these 21 neurons were tonically more (6 neurons) or less (2 neurons) active in water vs quinine blocks, during ITIs (*Figure 6C,D*). These neurons also showed a positive correlation in their response to the two different punishments, albeit with a weaker response to quinine (*Figure 6—figure supplement 2*). This could be due to the longer timecourse or smaller magnitude of the aversiveness of quinine compared to air puffs. Across both experiments, 20 of 50 identified serotonergic neurons showed tonic differences in firing for reward vs punishment blocks of trials. Thus, background firing rate in serotonergic neurons signal value across different magnitudes of reward and different types of punishment.

## Serotonergic neuron CS responses signal value

Next, we asked whether serotonergic neurons were value-coding within trials. We compared the firing rates during the CS, rather than US, for two reasons. First, there was no US during neutral trials. Second, outcomes were predicted by the CS, therefore neuronal responses to the US could be confounded by expectation. For each of the 23 serotonergic neurons with sufficient data for this analysis, the CS-induced firing rate as a function of value was monotonic (*Figure 7A,B*). As a

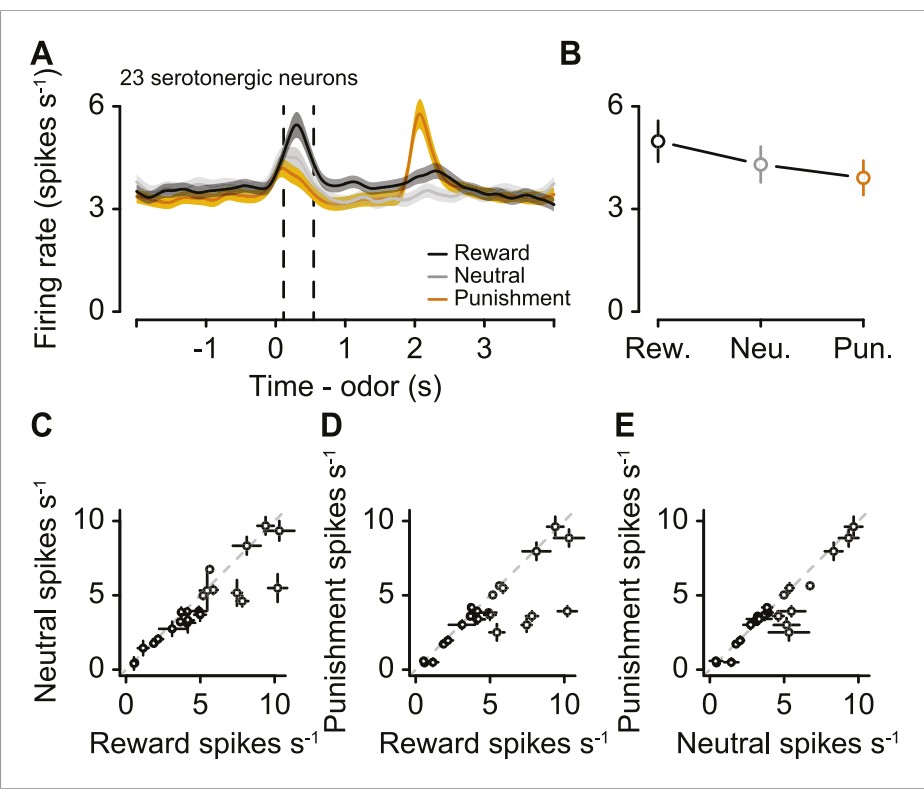

**Figure 7**. Serotonergic neurons signal value, not salience, in response to the CS. (**A**) Mean ± S.E.M. firing rate of serotonergic neurons during all three trial conditions. (**B**) Mean ± S.E.M. firing rate during the CS epoch (region bounded by dashed lines in **A**) for each trial condition. (**C**–**E**) Mean ±95% confidence intervals around firing rates during the CS epoch for neutral vs reward (**C**), punishment vs reward (**D**), and punishment vs neutral (**E**).

population, the CS firing rate was larger for reward than punishment (paired $t$-test, $t_{22} = 2.6$, p < 0.05), larger for reward than neutral ($t_{22} = 2.9$, p < 0.05), and larger for neutral than punishment ($t_{22} = 2.2$, p < 0.05; *Figure 7C–E*). This suggests that firing rate differences between reward and punishment trials reflected the value, not the salience, of those stimuli.

## Unidentified neuron responses

As with identified serotonergic neurons, we observed many unidentified neurons with firing-rate fluctuations from block to block (*Figure 8A,B*). We note that this population of unidentified neurons likely contains serotonergic as well as non-serotonergic neurons because of incomplete ChR2 expression or our strict criteria for identification.

We calculated the firing rate in the 2 s before the start of each trial during reward blocks and punishment blocks. 29 of 120 unidentified neurons showed significantly different pre-trial firing rates between reward and punishment blocks: 10 were more excited during reward blocks, 19 were more excited during punishment blocks (*Figure 8C,D*, Wilcoxon rank sum tests, p < 0.05).

In addition to these slow firing-rate fluctuations across minutes, 92 of 120 unidentified neurons showed task-related responses during the trial (ANOVA, all p < 0.01; *Figure 8—figure supplement 1*). These neurons were either excited or inhibited by rewards, punishments, reward-predicting cues, and punishment-predicting cues to varying degrees and durations. Although it is likely that there were false negatives (serotonergic neurons contained in the populations of unidentified neurons; *Figure 8—figure supplement 1*), there were significant differences between serotonergic and unidentified neurons. For instance, the duration of the significant response (excitation or inhibition) was significantly longer for unidentified neurons than for serotonergic neurons. That is, serotonergic neuron responses during the trial epoch (from CS to US) tended to be more phasic than those of the 92 task-responsive unidentified neurons (Fisher's exact test on the proportion of 100-ms bins significantly different from baseline, odds ratio = 0.643, p < 0.01). Finally, as has been observed previously in anesthetized cats and rats, we found both serotonergic and unidentified neurons that displayed 'clock-' and 'non-clock-like' firing patterns (*Figure 8—figure supplement 2B*; *Nakahama et al., 1981*; *Schweimer and Ungless, 2010*).

## Spike waveform properties of serotonergic neurons

Serotonergic neurons are typically identified extracellularly based on their wide spike shapes and low firing rates (*Bramwell, 1974*; *Aghajanian et al., 1978*; *Baraban et al., 1978*; *Baraban and Aghajanian, 1980*; *Aghajanian and Vandermaelen, 1982*; *Gallager, 1982*; *Heym et al., 1982*; *Wang and Aghajanian, 1982*; *Chiang and Pan, 1985*; *Fornal et al., 1985*, *1987*; *Cunningham and Lakoski, 1988*; *Levine and Jacobs, 1992*; *Ceci et al., 1994*; *Guzmán-Marín et al., 2000*; *Celada et al., 2001*; *Sakai and Crochet, 2001*; *Waterhouse et al., 2004*; *Miyazaki et al., 2011a*). These criteria have recently been called into question, however (*Park, 1987*; *Allers and Sharp, 2003*; *Kirby et al., 2003*; *Marinelli et al., 2004*; *Kocsis et al., 2006*; *Urbain et al., 2006*; *Hajós et al., 2007*; *Ranade and Mainen, 2009*; *Bromberg-Martin et al., 2010*; *Schweimer and Ungless, 2010*; *Schweimer et al., 2011*; *Gocho et al., 2013*; *Li et al., 2013*). We asked whether we could have identified serotonergic neurons in our data set in this way. Serotonergic neurons had significantly longer spike duration than unidentified neurons (Wilcoxon rank sum test, p < 0.05), although there was significant overlap in the distributions (*Figure 8—figure supplement 2A*). There was no significant difference between the mean firing rate (*Figure 8—figure supplement 2A*; Wilcoxon rank sum test, p > 0.6) or coefficient of variation of the inter-spike interval distributions (Wilcoxon rank sum test, p > 0.1) between serotonergic and unidentified neurons. Neither an expectation-maximization nor a k-means clustering algorithm could classify serotonergic neurons based on spike duration, firing rate, or shape of the inter-spike interval distribution.

## Discussion

By recording from identified serotonergic neurons, we showed that (1) a large fraction of serotonergic neurons showed tonic firing modulation depending on state value on long timescales (tens of seconds to minutes); (2) serotonergic neurons phasically responded (mostly by excitation) to punishments; and (3) a subset of serotonergic neurons was phasically excited by reward-predicting cues. These observations showed that serotonergic neurons signal reward and punishment on multiple timescales.

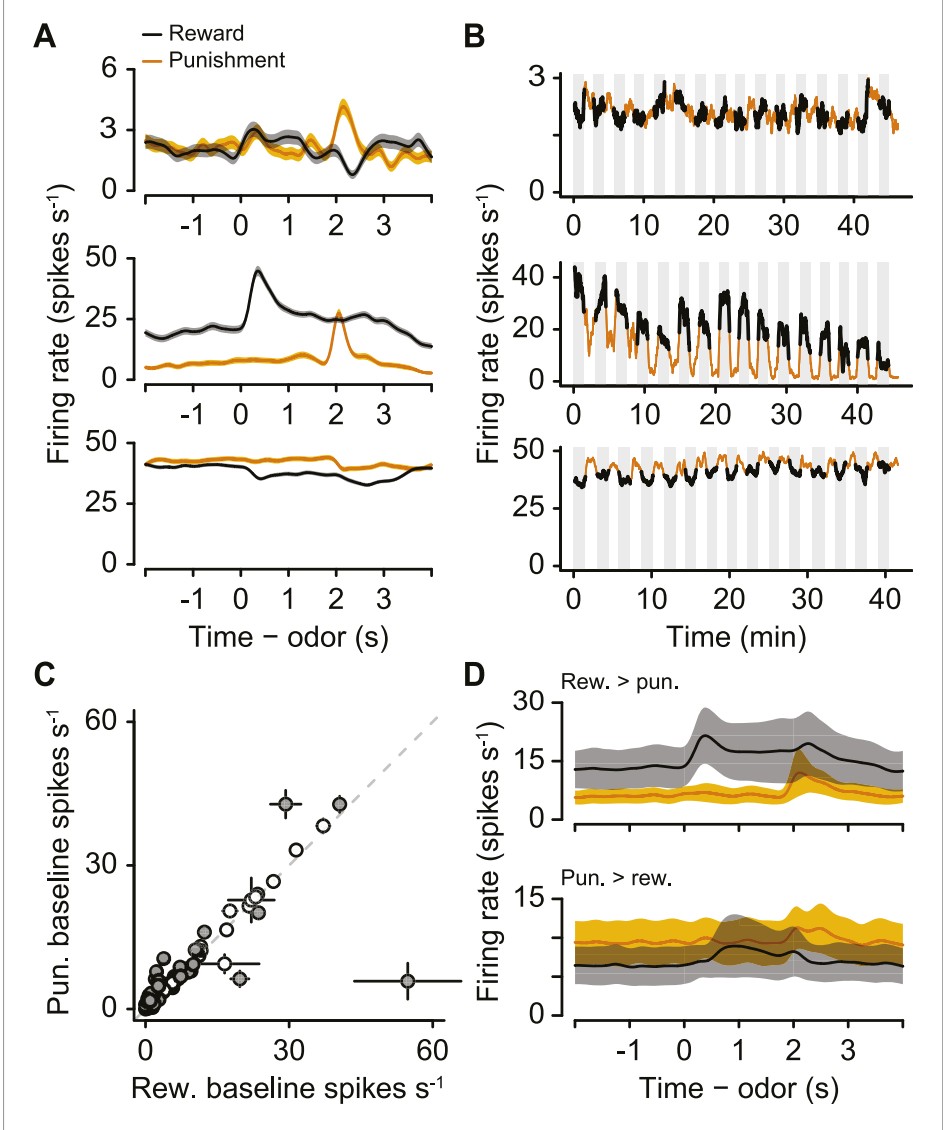

**Figure 8**. A population of unidentified neurons is more or less active during blocks of reward trials than punishment trials. (**A**) Mean ± S.E.M. firing rates of three example neurons during reward trials (black) and punishment trials (orange). Note the higher pre-trial firing rate during reward trials than punishment trials in the bottom two neurons. (**B**) Firing rate of the same three neurons across the timecourse of the experiment. Note the slow (across minutes) fluctuations in firing rate in the bottom two neurons that correlated with block type (reward: black, shaded regions; punishment: orange). (**C**) Mean ±95% confidence intervals around firing rates during the baseline epoch for punishment vs reward blocks for each unidentified neuron (significant data points are filled). (**D**) Mean ± S.E.M. firing rates of unidentified neurons with significantly higher baseline firing rates during reward (top) or punishment (bottom) blocks.

The following figure supplements are available for figure 8:

**Figure supplement 1**. Response profiles across all neurons and all trial types.

**Figure supplement 2**. Serotonergic neurons cannot be identified based on firing properties in this data set.

---

The dorsal raphe nucleus contains diverse types of neurons (*Marinelli et al., 2004*; *Commons, 2009*; *Fu et al., 2010*; *Hioki et al., 2010*). Although previous recording studies have used indirect methods based on spike shape or firing rate measures (*Fornal et al., 1996*; *Miyazaki et al., 2011a*), it

has been difficult to separate the responses of serotonergic from non-serotonergic raphe neurons during behavioral tasks. Indeed, our data set of identified serotonergic neurons contained diverse waveforms and firing properties, making it critical to use a more direct method to identify serotonergic neurons. We observed some diversity in the responses of identified serotonergic neurons (cf. *Ranade and Mainen, 2009*) but, to describe the full diversity of serotonergic neurons, a larger sample size is required. We thus focused on salient features that were shared by a large fraction of identified serotonergic neurons, which we observed in our sample size of 50 identified serotonergic neurons. In the following, we discuss potential implications of our findings in the context of theories of behavioral regulation and serotonin signaling.

## Tonic firing of serotonergic neurons tracks state values

Reward and punishment can exert their effects on behavior on multiple timescales. It has been proposed that the average reward rate on relatively long timescales (or state value) regulates the vigor of behavioral responding in a manner relatively non-specific to actions (*Niv et al., 2006*; *Wang et al., 2013*). However, the neural mechanisms that regulate this process remain unclear. Much attention has been paid to a potential involvement of tonic dopaminergic firing in this process (*Niv et al., 2006*; *Cools et al., 2011*). In contrast to this proposal, we did not find that dopaminergic neurons changed their baseline firing according to state values. Our data showed, instead, that 40% of serotonergic neurons changed their baseline firing depending on the state value of the environment. We observed this effect using rewards and punishments of the same sensory modality, and using brief punishments (air puffs), in which the acute aversiveness likely did not persist into the ITI. These firing rate changes were relatively small in magnitude (around 1–2 spikes s$^{-1}$), but given the low baseline firing rates of serotonergic neurons, these changes corresponded to around 20–100% increases in firing rates, which could have led to substantially higher serotonin release.

Our data also showed that serotonergic neurons exhibited transient activations associated with various task events (*Nakamura et al., 2008*; *Ranade and Mainen, 2009*). Serotonergic neuron responses to reward- or punishment-predictive cues as well as reward or punishment were transient, typically lasting less than 500 ms, and relatively small in magnitude (5–10 spikes s$^{-1}$), in agreement with previous studies (*Nakamura et al., 2008*; *Ranade and Mainen, 2009*; *Bromberg-Martin et al., 2010*; *Miyazaki et al., 2011a*; *Inaba et al., 2013*), though substantially smaller than a recent one (*Li et al., 2013*). Previous studies observed that dorsal raphe neurons exhibited sustained activities that appeared to track moment-to-moment changes in value triggered by sensory cues and outcomes, within a trial (*Nakamura et al., 2008*; *Bromberg-Martin et al., 2010*). Although these activities lasted for several hundreds of milliseconds to seconds, our identified serotonergic neurons showed relatively transient activities within trials. In contrast, our data showed that many unidentified neurons showed sustained activities within trials, suggesting that sustained activities may be more common in non-serotonergic neurons. Another recent set of studies found that putative serotonergic neurons showed firing modulations lasting up to several seconds, during a task in which rats waited for a reward (*Miyazaki et al., 2011a*), and that manipulations of serotonergic signaling altered waiting behavior (*Miyazaki et al., 2014*; *Fonseca et al., 2015*). This raises the interesting possibility that both time and reward value modulate the firing of serotonergic neurons, increasing the flexibility of the serotonergic signal. Indeed, the tonic signal we observed could be the subjective value of waiting (i.e., waiting for punishments elicits a low-value state, whereas waiting for rewards elicits a high-value state). A third study found that dorsal raphe neurons, on average, fired at a higher rate before cues that predicted rewards relative to cues that predicted no reward (*Li et al., 2013*). In this work, trials were also delivered in a block-wise fashion, though it was not possible to identify serotonergic neurons, nor was there an analysis of the slow modulations in firing rate in individual neurons.

The sign of value-dependent changes in tonic firing varied across serotonergic neurons: some increased and others decreased during periods of high state values although, on average, high-value blocks were associated with higher tonic firing rates. Nevertheless, the topography of projections from the raphe (*Imai et al., 1986*; *Vertes, 1991*; *Lowry et al., 2000*; *Chandler et al., 2013*) and the physiological topography within the raphe (*Lowry et al., 2000*; *Crawford et al., 2010*) suggest the potential for a specific mapping between subsets of serotonergic neurons and their functions. Our results cannot distinguish these subpopulations of serotonergic neurons. It remains to be examined whether different firing patterns of serotonergic neurons correspond to specific subpopulations.

Together, these results suggest that background serotonin could serve as an explicit signal of state values. It is important to note that our data demonstrate a relative value code during the task (blocks of trials of different values), but do not speak to the possibility that serotonergic neurons signal an absolute state value (cf. *Figure 3—figure supplement 3*). As discussed above, reward and punishment affect behavior on multiple timescales. Our finding that tonic serotonergic firing tracks values over long timescales raises the possibility that tonic serotonin regulates long-lasting affective states.

## Dopamine and serotonin do not signal opposite information

In addition to the changes in tonic firing, our data showed that a large fraction of serotonergic neurons were excited by reward-predictive cues and unpredicted reward on shorter timescales (i.e., within trials). Although there was substantial diversity in serotonergic neuron firing patterns, during reward trials, on average, their response to reward-predicting cues resembled those of dopaminergic neurons signaling reward prediction error (*Schultz et al., 1997*; *Cohen et al., 2012*). It is important to note, however, that the magnitude of increases was smaller, and did not appear to signal prediction errors for most neurons (*Figures 3D,H, 4I*). In addition, we found that many serotonergic neurons were excited by punishments. It is important to disentangle whether the phasic responses to air puffs we observed are due to aversiveness or other factors such as saliency or relief from punishment (*Heym et al., 1982*; *Waterhouse et al., 2004*; *Dayan and Huys, 2009*; *Cools et al., 2011*). The short response latency suggests that it is unlikely to be relief from punishment, but the nature of phasic air puff responses remains to be clarified. Of course, all of these signals could be expressed by the population of serotonergic neurons. In addition, it will be important to understand why the phasic response to air puffs was stronger than the phasic response to air-puff-predicting cues.

It has been proposed that serotonergic and dopaminergic neurons work largely in an opponent manner (*Daw et al., 2002*). This idea was supported by pharmacology and intracranial self-stimulation (*Redgrave, 1978*), as well as a few recording studies that suggested phasic activations of serotonergic neurons by aversive stimuli (*Aghajanian et al., 1978*; *Montagne-Clavel et al., 1995*; *Schweimer and Ungless, 2010*). However, in these studies, responses were measured in anesthetized animals or the effect of reward was not examined, leaving unanswered whether phasic serotonin signals purely aversive information. Our data showed that many single serotonin neurons were activated by both reward-predictive cues and punishment (*Figure 5C*). Although it remains unclear whether these phasic firing patterns can be unified to encode a particular variable, or whether the response to punishment was due to its sensory nature or the relief arriving at the end of the punishment, these firing patterns appear to be inconsistent with the idea that serotonergic neurons send an opponent signal compared to dopaminergic neurons, though median raphe serotonergic neurons could provide such a signal (*Daw et al., 2002*). This lack of pure opponency is consistent with recent studies measuring putative serotonergic firing and serotonin concentration in a waiting task (*Miyazaki et al., 2011a*, *2011b*).

## Multiplexing of signals on multiple timescales

Our results suggest that serotonergic neurons multiplex information about reward and punishment on multiple timescales. We propose that slow, value-related firing could represent state value, whereas phasic responses to CS or US could encode a different variable. This slow signal appears to require some time to change (*Figure 3—figure supplement 2*). Future studies should clarify how downstream neurons read out tonic vs phasic serotonin signals. It is possible that tonic serotonin has very different effects at target neurons and on behavior than phasic serotonin due to receptors with different affinities or other mechanisms (*Daw et al., 2002*). In addition, serotonergic neurons are known to contain other transmitters (*Varga et al., 2009*; *Liu et al., 2014*), suggesting that the slow and fast timescales could correspond to the action of different transmitters on target neurons, or to downstream circuit effects with differing durations. The function of serotonin could be target-dependent (*Deakin and Graeff, 1991*), timing-dependent (*Daw et al., 2002*; *Dayan and Huys, 2009*), or dependent on co-release of other transmitters (*Dayan and Huys, 2009*; *Varga et al., 2009*; *Liu et al., 2014*; *McDevitt et al., 2014*). Another possibility is that serotonin combines with other circuits to form logical combinations postsynaptically (for example, serotonin AND dopamine codes for reward, whereas serotonin AND NOT dopamine codes for punishment).

How do serotonergic neurons compute the signals over short (hundreds of milliseconds) and long (minutes) timescales? Serotonin release is controlled by many types of afferents. The densest include

frontal cortex, basal forebrain areas (bed nucleus of the stria terminalis, substantia innominata, ventral pallidum), hypothalamic nuclei (preoptic nucleus, lateral and posterior nuclei), lateral habenula, and several midbrain and brainstem structures (*Peyron et al., 1998*; *Pollak Dorocic et al., 2014*; *Ogawa et al., 2014*; *Weissbourd et al., 2014*). Neurons in the VTA and substantia nigra pars compacta (SNc), some dopaminergic, provide input to raphe serotonergic neurons (*Beckstead et al., 1979*; *Pollak Dorocic et al., 2014*; *Ogawa et al., 2014*; *Weissbourd et al., 2014*). These neurons are phasically excited by rewards or reward-predicting cues (*Schultz et al., 1997*; *Cohen et al., 2012*), but their activity appears not to correlate with longer-term changes in value (*Figure 3* above; *Matsumoto and Hikosaka, 2009*). They may excite serotonergic neurons via D2 receptors (*Haj-Dahmane, 2001*; *Aman et al., 2007*; but see; *Martín-Ruiz et al., 2001*). Interestingly, dopaminergic stimulation of putative serotonergic neurons in vitro caused excitation that persisted beyond the application of dopamine (*Haj-Dahmane, 2001*). It is possible that serotonergic neurons could accumulate information about rewards via short pulses of input from dopaminergic neurons, although our observation of shorter reward-predicting CS response latencies in serotonergic vs dopaminergic neurons suggests the opposite.

## Spike waveform properties of serotonergic neurons

There are several possible explanations for why we could not use spike width to clearly identify serotonergic neurons. First, unidentified neurons likely contained serotonergic neurons that we could not identify using our stringent criteria. Second, the shape of extracellulary-recorded spikes depends on the position of the electrode tip to the soma and dendrites of the recorded neuron (*Schultz, 1986*; *Harris et al., 2000*; *Henze et al., 2000*; *Buzsáki et al., 2012*). Given that the geometry of dendritic and axonal processes in the dorsal raphe is diverse, with fusiform, multipolar, and ovoid somata (*Diaz-Cintra et al., 1981*), it is potentially difficult to optimize the location of electrode tip relative to cell bodies across the population (cf. *Schultz, 1986*). Indeed, we did not attempt to optimize this position, once we found an identified serotonergic neuron. Third, our nichrome wires could have introduced recording bias away from small cells or certain cell shapes, potentially less of an issue for glass pipette recordings (*Towe and Harding, 1970*; *Shoham et al., 2006*; *O'Connor et al., 2010*). Indeed, in our previous study in the ventral tegmental area, using the same type of electrode, we did not find the oft-reported difference between spike width for dopaminergic vs non-dopaminergic neurons (*Cohen et al., 2012*). Careful simultaneous intra- and extracellular measurements (*Harris et al., 2000*; *Henze et al., 2000*) in the dorsal raphe are needed to resolve this.

## Summary

The present study revealed that serotonergic neurons can use tonic as well as phasic firing to convey reward information. This raises questions as to whether these different modes of firing convey distinct information, whether they have different impacts on target neurons (notably, on dopaminergic neurons [*Fibiger and Miller, 1977*; *Watabe-Uchida et al., 2012*; *Ogawa et al., 2014*]), and how they are calculated. Although our data likely do not reveal the full diversity of serotonergic neuron firing dynamics, our results suggest distinguishing these distinct modes is crucial to teasing apart the seemingly complex functions of serotonin in various brain functions and disorders.

# Materials and methods

## Animals

For dorsal raphe recordings, we used nine adult male mice, backcrossed with C57BL/6J mice, heterozygous for Cre recombinase under the control of the serotonin transporter gene (Slc6a4$^{tm1(cre)Xz}$; *Zhuang et al., 2005*). For VTA recordings, all of which came from the task in *Figure 1*, we used four adult male mice backcrossed with C57BL/6J mice, heterozygous for Cre recombinase under the control of either the dopamine transporter (3 mice) or tyrosine hydroxylase gene (1 mouse) (Slc6a3$^{tm1.1(cre)Bkmn}$/J and B6.Cg-Tg(Th-cre)$^{1tmd}$/J, respectively, The Jackson Laboratory; *Savitt et al., 2005*; *Bäckman et al., 2006*). We did not observe any differences between the different genotypes during the behavioral task. For cell counting, we used three additional adult male *Sert*-Cre mice. Animals were housed on a 12 hr dark/12 hr light cycle (dark from 06:00–18:00) and each performed the conditioning task at the same time of day, between 07:00 and 19:00. All surgical and experimental procedures were in accordance with the National Institutes of Health Guide for the Care and Use of Laboratory Animals and approved by the Harvard or Johns Hopkins Institutional Animal Care and Use Committees.

## Surgery and viral injections

Mice were surgically implanted with a head plate and a microdrive containing electrodes and an optical fiber. During a prior surgery, we injected 200–500 nl adeno-associated virus (AAV), serotype 2/5, using the EF1α promoter, carrying an inverted ChR2 (H134R)-EYFP flanked by double loxP sites (*Nagel et al., 2003*; *Boyden et al., 2005*; *Atasoy et al., 2008*) into the dorsal raphe stereotactically (from bregma: 4.4–4.7 mm posterior, 0.1–0.2 mm lateral, 1.9–2.3 mm ventral), or into VTA as previously described (*Cohen et al., 2012*).

All surgery was performed under aseptic conditions with animals under ketamine/medetomidine (60 and 0.5 mg/kg, I.P., respectively) or isoflurane (1–2% at 0.5–1.0 l/min) anesthesia. Analgesia (ketoprofen, 5 mg/kg, I.P.; buprenorphine, 0.1 mg/kg, I.P.) was administered postoperatively.

## Behavioral task

After at least 1 week of recovery, mice were water-deprived in their home cage. Weight was maintained within 90% of their full body weight. Animals were head-restrained using a custom-made metal plate and habituated for 1–2 day while head-restrained before training on the task. Odors were delivered with a custom-made olfactometer (*Uchida and Mainen, 2003*). Each odor was dissolved in paraffin oil at 1/10 dilution. 30 µl of diluted odor was placed inside a filter-paper housing. Odors were isoamyl acetate, 1-butanol, N-citral, eugenol, (+) limonene, (−) carvone, (+) carvone, allyl tiglate, eucalyptol, acetophenone, hydroxymethylpentylcyclohexenecarboxaldehyde, 3-hexanone, pentyl acetate, 1-hexanol, p-cymene, and ethyl butyrate, and differed for different animals. Odorized air was further diluted with filtered air by 1:10 to produce a 500 ml/min total flow rate. Licks were detected by breaks of an infrared beam placed in front of the water tube.

We delivered one of three odors, selected pseudorandomly, for 1 s, followed by a delay of 1 s and an outcome. Each odor predicted a different outcome: a drop of water (4 µl), no outcome, or an air puff delivered to the animal's face. ITIs were drawn from an exponential distribution with a rate parameter of 10 (i.e., $P(ITI) = 1/10 \exp(-x/10)$). This resulted in a flat ITI hazard function, ensuring that expectation about the start time of the next trial did not change over time. A 15 kHz tone lasting 1 s signaled block changes, ending 1 s before the start of the next trial. Data were obtained from 141 sessions (19–28 sessions per animal). For 17 identified serotonergic neurons, we omitted rewards during 10% of big-reward trials. Animals performed between 400 and 700 trials per day (533 ± 120 trials, mean ± SD). Free rewards were delivered before the start of the conditioning task. We did not observe differences in lick rate (*Figure 1*) or sniff rate (counted manually during 1-s intervals of 20 reward trials and 20 punishment trials for two mice; t-tests, $t_{38} = 1.28$, $p > 0.20$, $t_{38} = 1.05$, $p > 0.29$ for each mouse) during ITIs of different block types.

For 16 of 29 serotonergic neurons, block type varied randomly, whereas for 13 of 29 serotonergic neurons, and all dopaminergic neurons, block type alternated between reward and punishment. For 16 of 29 serotonergic neurons, neutral blocks were included, in which case an additional odor was used as a CS. For 2 of 29 serotonergic neurons, block size was 20 trials. We trained each animal on the task with randomly-interleaved trials for 5 days before beginning the blocked structure.

For the task including quinine as a US, we used water (4 µl) or chocolate milk (1%, Hood, Lynnfield, Massachusetts, 4 µl) as a reward US and quinine HCl (0.5–1.0 mM, 4 µl) as a punishment US. To ensure that animals ingested the quinine, which is known to be aversive (*Schoenbaum et al., 1998*; *Berridge, 2000*; *Peciña and Berridge, 2005*; *Roitman et al., 2005*), we placed the delivery tube at the entry to their mouths. Other task parameters were the same as above.

For the freely-moving behavioral task, we trained five adult male C57BL/6J mice to perform a two-alternative forced choice task as follows (*Figure 1—figure supplement 1*). An odor cue (1-hexanol) delivered at a central port was a start cue that signaled the mouse to choose between one peripheral port that contained water (4 µl) and a second that contained water (4 µl) and an air puff. After an ITI (same distribution as for the tasks described above), the next trial began. Mice performed 300 trials of this task for 5 days, after 6 days of shaping and training.

## Electrophysiology

We recorded extracellularly from multiple neurons simultaneously using custom-built 200-µm-fiber-optic-coupled screw-driven microdrives with eight implanted tetrodes (four nichrome wires wound together, Sandvik, Palm Coast, Florida). Tetrodes were glued to fiber optics with epoxy or

cyanoacrylate. The ends of tetrodes were 400–600 µm from the ends of fiber optics. Neural signals and time stamps for behavior were recorded using DigiLynx recording systems (Neuralynx, Bozeman, Montana). Broadband signals from each wire were filtered between 0.1 and 9000 Hz and recorded at 32 kHz. To extract the timing of spikes, signals were bandpass-filtered between 300 and 6000 Hz and sorted online and offline.

To verify that our recordings targeted serotonergic or dopaminergic neurons, we used ChR2 to observe stimulation-locked spikes (*Cardin et al., 2009*; *Lima et al., 2009*; *Cohen et al., 2012*). The optical fiber was coupled to a diode-pumped solid-state laser with analog amplitude modulation (Laserglow Technologies, Toronto, Canada). For each neuron, we delivered trains of 10 light pulses, each 5 ms long, at 1, 5, 10, 20, and 50 Hz at 473 nm at 5–20 mW/mm$^2$, before and after the experimental session. Higher intensities typically resulted in light-evoked spike waveforms that did not match spontaneous ones. Therefore, we adjusted the light intensity after observing the responses at the beginning of experiments. The increasing latency of light-evoked spiking as a function of stimulation frequency indicates that the response was not due to photochemical artifact (*Figure 2E*). Spike shape was measured using a broadband signal (0.1–9000 Hz) sampled at 32 kHz.

We used two criteria to include a neuron in our data set. First, the neuron must have been recorded within 500 µm of an identified serotonergic neuron, to ensure that all neurons came from the dorsal raphe (except for dopaminergic neurons, which all came from VTA). Second, the neuron must have been well isolated. To measure isolation quality, we calculated the L-ratio (*Schmitzer-Torbert and Redish, 2004*), which approximates the fraction of 'contaminated' spikes. Smaller L-ratios indicate better isolation. All neurons in the data set had L-ratios < 0.05 and signal-to-noise ratios of > 5 dB. Identified serotonergic neurons, of which there were 50 across experiments, came from all nine mice (a range of 1–10 per mouse).

## Data analysis

To measure firing rates, peristimulus time histograms (PSTHs) were constructed using 10-ms bins. To calculate spike density functions, PSTHs were convolved with a function resembling a postsynaptic potential, $(1 - \exp(-t)) \exp(-t/200)$, for time $t$. For display (but not analysis), we smoothed spike density functions with a smoothing spline with 30 degrees of freedom (*Kass et al., 2005*). To determine whether a neuron showed a significant task-related response, we calculated an ANOVA on the trial-by-trial firing rates during the baseline period (1 s before odor onset), CS period (from odor onset to odor offset), delay (from odor offset to outcome onset), and US period (from outcome onset to 500 ms after outcome onset). The factors were task epoch (baseline, CS, delay, or US) and outcome type. Normality was tested by Kolmogorov–Smirnov tests and quantile–quantile plots. All two-group comparisons were two-sided. Effect sizes for each experiment were determined post-hoc using Cohen's U$_3$ (cf. *Hentschke and Stüttgen, 2011*).

Light-evoked spikes were detected during the 10 ms after light onset. If less than 20% of light pulses evoked a spike (defined as a waveform that matched that of the isolated unit) during the 10 ms after light onset (upper left points in *Figure 2B*), the maximum absolute voltage during that interval was used as the light-evoked 'response'. Euclidean distances between spontaneous and light-evoked spike waveforms were calculated by aligning the larger of the positive or negative peak of each waveform, averaging separately, and aligning the peaks of the averages. The distance was calculated using the full duration of the spontaneous spike (spike duration was measured as the first time until the last time at which the voltage was significantly different from baseline using Wilcoxon rank sum tests). The energy of the light-evoked waveform is defined as the integral of the squared voltage values ($\int v^2 dt$).

ROC curves were calculated by comparing the distribution of firing rates (spike density functions) across trials in 100-ms bins to the distribution of baseline firing rates (1 s before odor onset, using 100-ms bins). The duration of significant responses were calculated using Wilcoxon rank sum tests comparing the baseline firing rate to the firing rate in the response window of interest, bin by bin. The number of bins in which $p < 0.05$ was taken as the duration of the significant response (after Bonferroni corrections).

Expectation-maximization clustering was performed using hierarchical clustering for parameterized Gaussian mixture models, setting the number of clusters to 2 (one cluster of 'identified serotonergic neurons' and one of 'unidentified neurons'), with model selection by Bayesian Information Criterion.

All statistical tests were done with Bonferroni corrections for multiple comparisons. Analyses were done with R (http://www.r-project.org/).

## Immunohistochemistry

After recording, which lasted between 19 and 28 days, mice were given an overdose of ketamine/ medetomidine, exsanguinated with saline, perfused with 4% paraformaldehyde, and brains were cut in 50–100 µm coronal sections. Sections were immunostained overnight with a primary antibody to 5-HT (1:200; S5545, Sigma–Aldrich, St. Louis, Missouri; *Steinbusch et al., 1978*; *Bras et al., 1986*; *Dai et al., 2008*), then incubated with an Alexa594-coupled secondary antibody for 2 hr (1:200; Invitrogen, Carlsbad, California). Sections were further counterstained with 4′,6-diamidino-2-phenylindole (DAPI) to visualize nuclei. Recording sites were identified and verified to be amid EYFP expression and 5-HT staining in the dorsal raphe.

## Acknowledgements

We thank V Murthy, N Eshel, M Watabe-Uchida, D O'Connor, R Heitz, and members of the Uchida lab for discussions, C Dulac and V Murthy for sharing resources, K Deisseroth for the AAV-FLEX-ChR2-EYFP construct, the UNC Vector Core Facility for viruses, X Zhuang for the *Sert*-Cre mouse, and E Soucy and J Greenwood for technical support. This work was supported by a Howard Hughes Medical Institute Fellowship from the Helen Hay Whitney Foundation (JYC); an NSF Graduate Research Fellowship (MWA); and NIH (R01MH095953, R01MH101207) (NU).

## Additional information

### Funding

| Funder | Grant reference | Author |
|---|---|---|
| Helen Hay Whitney Foundation (HHWF) | HHMI Fellowship from the HHWF | Jeremiah Y Cohen |
| Howard Hughes Medical Institute (HHMI) | HHMI Fellowship from the HHWF | Jeremiah Y Cohen |
| National Science Foundation (NSF) | Graduate Research Fellowship | Mackenzie W Amoroso |
| National Institutes of Health (NIH) | R01MH095953 | Naoshige Uchida |
| National Institutes of Health (NIH) | R01MH101207 | Naoshige Uchida |

The funders had no role in study design, data collection and interpretation, or the decision to submit the work for publication.

### Author contributions

JYC, Conception and design, Acquisition of data, Analysis and interpretation of data, Drafting or revising the article; MWA, Acquisition of data, Analysis and interpretation of data; NU, Conception and design, Drafting or revising the article

### Author ORCIDs

Mackenzie W Amoroso, http://orcid.org/0000-0001-7368-4456

### Ethics

Animal experimentation: All surgical and experimental procedures were in accordance with the National Institutes of Health Guide for the Care and Use of Laboratory Animals and approved by the Harvard or Johns Hopkins Institutional Animal Care and Use Committees.

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
