## [Decision Letter]

Thank you for sending your work entitled “Serotonergic neurons signal reward and punishment on multiple timescales” for consideration at *eLife*. Your article has been favorably evaluated by Timothy Behrens (Senior editor and Reviewing editor) and 3 reviewers, one of whom, Geoffrey Schoenbaum (Reviewer #1), has agreed to reveal his identity.

The Reviewing editor and the reviewers discussed their comments before we reached this decision, and we would like to invite a revised submission in line with the review comments below.

*Reviewer #1*:

Here the authors record from optogenetically-identified SERT+ neurons in mice performing a simple odor-based discrimination task in which different odors predicted reward, punishment, or very low probability of reward (termed neutral). Trials were arranged in blocks in order to modulate the motivation, state value or mood of the mouse. Mice were shown to discriminate in their licking behavior on the different trial types. Subsequent analyses of SERT+ neurons (29 in the main task and smaller subpopulations in various follow up studies for a grand total of ∼50) showed that they exhibited correlates based on value (not salience or sensory features) both within trials during CS and US periods and between trials based on block. Activity was compared to that of TH+ neurons recorded I think in a previously published study. Activity was superficially similar in some regards during trials, but differed in details and also during the periods between trials such that the authors concluded that the two populations were doing different things in both periods. Overall the authors conclude that the SERT+ neuron activity correlates with predicted value or state value on both long time scales (trial blocks) and short time scales (within trials) and that this is the most salient feature of their activity.

Overall the paper is excellent. The study is simple and the analyses are generally clearly and carefully presented, logical, and easy to follow. The underlying question is important and the task design and analysis is appropriate to answer it. Indeed the results are extremely interesting to me and I think will be to the field at large. Of particular value is the comparison to TH+ neurons and also the comparison across time scales. I do not think this has been done clearly before. It is quite clever and has yielded interesting and informative data. I also enjoyed and found valuable the discussion, which did a very nice job relating the current findings both to several influential but largely theoretical proposals for serotonergic function and to the relatively sparse existing single unit data. The authors nicely link their results to those from Doya and colleagues on firing during delays.

Of course I do have a few suggestions and questions for the authors. My only serious concern probably is that as far as I can tell the authors have little or no evidence that the block design resulted in changes in motivation or mood. While the licking data obviously shows the mice understood the meaning of the odor cues, it does not show that they were attending to the blocks. While I think it is highly likely they were, especially if they were well-trained (data?), it would strengthen the claims to provide some behavioral data on this. Differences in trial initiation speed, proportions of trials completed would be sufficient I think. If this is not possible, I guess it does not negate the significance of the neural correlates, but I think it is a significant gap that should be filled in if possible. Likewise it would be useful to have behavior from the various follow up tasks shown. These data are an important part of the report, especially in showing that the activity is not salience or simply sensory.

I also have various other questions, which I will list in no particular order:

1) The first paragraph of the Discussion seems to be about performance, but much of the data/ideas about dopamine and even serotonin's influences on behavior is due to learning. I found this a bit confusing, especially as learning is then lumped in with this function in the second paragraph.

2) Please clarify where the dopamine neuron data came from? Is this from the prior published study? Is the task identical to that used here? I was a bit confused.

3) It would be helpful if the authors would link the individual figure panels a bit more closely to their text. At times I felt like I was reading two papers in parallel, especially as the figure panels are not well enough labeled as to not require reading the captions.

4) In the sliding window ROC analysis, the authors find that 28/29 SERT+ neurons are phasically activated. What does this mean? Meaning that they had 1 bin above chance across the analysis? How many bins and what p level is this?

5) In the Results, there is a paragraph describing the response of the neurons to unexpected reward and punishment. It comes a bit out of the blue. I was not clear even where these data came from in the context of the task. I think this should be more clearly explained. I'd also like to see the activity directly compared to the expected. This will make more clear whether there is an implication of error signaling here or if it is just that the neurons are firing to the US the same way they are during the trials.

6) Related to point #5, I think that unless you find clear evidence that the neurons fire to unexpected reward or punishment differently from how they fire to expected, then you should not emphasize this point in the subsection headed “Dopamine and serotonin do not signal opposite information” in the Discussion. There you start by emphasizing the response. To me this carries the connotation of error signals. This has not been shown and Doya et al. have provided strong data against this. And I think the results here are largely in agreement. Assuming this is not the intent here, this might be modified. (If it is the intent, then I think much more must be done to show that these are error signals.)

*Reviewer #2*:

In this very interesting study, Cohen et al. provide what is the most comprehensive characterization to date of the activity of optogenetically-identified dorsal raphe serotonergic cells in a Pavlovian paradigm involving predictions of rewards and punishments. The results are quite complex, with differential modulations over multiple timescales, and not complete. However, they represent an important step forward.

Critical questions:

1) We need to know a bit more about the behaviour of the animals: do they orient in any way to all three types of cues, is this response related to the 5-HT activity? Can they avoid sampling the odour in the aversive blocks? Do they show other changed behavior? Does the licking response show anything like the topography seen in Figure 3—figure supplement 2? This is important given evidence from the likes of Barry Jacobs about activity-related influences on 5-HT firing.

2) I was confused by the status of the neutral trials. In the Methods, it sounds as if neutral stimuli were interspersed with probability 0.1 in each block (also by reading of Figure 1), in which case I'd like to see what happens on those trials. But later in the paper, it sounds as if there were whole blocks with neutral trials too. In the latter case, if these blocks were relatively unusual, then I think it hard to make any argument from them about salience versus value coding, since they could be salient by virtue of being rare.

3) I was disappointed at how little analysis there was of the data in Figure 6 when there was an additional level of reward. It is important to report what happens to other aspects of the activity of the 5-HT neurons, for instance, how are the prominent value-related CS responses affected? How does the monotonic relationship in Figure 6 relate to baseline activation by reward/punishment? The logic of these analyses is the same as the logic of the authors having changed reward values in the first place.

4) Schwiemer et al. reported different classes of physiological activity for their 5-HT neurons, and that the response to punishment was related (e.g. 'clock-like'). Was there any evidence of that here, or could that have arisen from the anaesthetic?

5) Was there any modulation of the baseline or CS activity as a function of the length of the ITI (it would be good to show an interquartile split)? This directly bears on the nature of prediction revealed by these neurons.

6) The Introduction is a bit woeful compared with the care in the rest of the paper. Tops et al. isn't correctly represented, the stimulation work on patience and preference and optogenetic activation of 5-HT neurons should be better recognized; likewise the heterogeneity revealed by cFOS imaging of the raphe (from Lowry, Maier and others). Also, there is not really a competition between tonic and phasic aspects of dopamine or serotonin signaling. These are different facets of the signal that could even be read by different downstream mechanisms. This comes across more reasonably in the Discussion.

7) The structure of the methods in the Results was a bit puzzling, not only the issue of neutral trials/blocks mentioned above, but also not thoroughly discussing the methods for the DA part of the study.

Reviewer #3:

This study from Cohen et al. investigates the phasic and tonic responses of optogenetically identified serotonergic neurons in the raphe nucleus to rewarding and aversive stimuli. They find that, unlike dopaminergic neurons, serotonergic neurons exhibit long-lasting tonic signals that track the value of the entire block. Serotonergic neurons also respond phasically to rewarding and punishing stimuli, with stronger phasic responses to the CS on rewarded blocks and to the US on punished blocks. Overall the experiments are well designed and executed, and the analyses are sound. However, several points require further explanation or clarification as detailed below: 1) The authors should clarify the methodology used to identify serotonergic neurons; namely the “light-evoked energy” analysis in Figure 2 should be explained more thoroughly, as it is not a standard method.

2) The authors should include more discussion about why the neural responses to punishment occur consistently later than those to reward. Why do they not respond phasically to the punishment-predicting cues but only the reward-predicting cues?

3) Figure 6 requires more explanation. The plot in panel (a) does not very convincingly convey the conclusions the authors state, and it is confusing that in panel (b) a larger reward is correlated with decreased activity in the serotonergic cells.

4) Are there differences in the anatomical location within the raphe of cells that respond preferentially to reward vs. punishment?

5) The authors should discuss their results in light of recently published work employing optogenetic stimulation of serotonergic neurons in different reward-related behavioral paradigms (e.g. [66] Neuron; [74] Cell Rep.).

---

## [Author Response]

Reviewer #1:

*My only serious concern probably is that as far as I can tell the authors have little or no evidence that the block design resulted in changes in motivation or mood. While the licking data obviously shows the mice understood the meaning of the odor cues, it does not show that they were attending to the blocks. While I think it is highly likely they were, especially if they were well-trained (data?), it would strengthen the claims to provide some behavioral data on this. Differences in trial initiation speed, proportions of trials completed would be sufficient I think. If this is not possible, I guess it does not negate the significance of the neural correlates, but I think it is a significant gap that should be filled in if possible. Likewise it would be useful to have behavior from the various follow up tasks shown. These data are an important part of the report, especially in showing that the activity is not salience or simply sensory*.

We thank the reviewer for this important point. We analyzed the behavioral data during the tone that signaled the start of a new block. During the 35 sessions in which block type predictably alternated between reward and punishment, mice licked significantly more during the tone (as analyzed in the subsection “Behavioral Task”) that indicated the start of a reward block versus the start of a punishment block (Wilcoxon signed rank test, p <0.01). Because this tone was the same for both reward-to-punishment and punishment-to-reward block transitions, this indicates that mice attended to the current block type. We added this to lines 106-110 and a new Figure 1—figure supplement 1.

We also analyzed the behavioral data from the other experiments, with the exception of the one with quinine, and added those to a new Figure 6—figure supplement 1. We could not measure licking during the experiment with quinine, because we had to place the delivery tube inside the mouth to ensure consumption.

*I also have various other questions, which I will list in no particular order*:

*1) The first paragraph of the Discussion seems to be about performance, but much of the data/ideas about dopamine and even serotonin's influences on behavior is due to learning. I found this a bit confusing, especially as learning is then lumped in with this function in the second paragraph*.

We thank the reviewer for this comment. We discuss theories about the relationship between dopamine and serotonin and learning in the Introduction. We focused on the link to performance in the Discussion because it is most relevant to our data.

*2) Please clarify where the dopamine neuron data came from? Is this from the prior published study? Is the task identical to that used here? I was a bit confused*.

We apologize for the confusion. These are new data, using the same task for serotonergic neurons, to ask whether dopaminergic neurons also show tonic value-related signals (interestingly, they did not). We removed the reference to our previous work to clarify that these are new data.

*3) It would be helpful if the authors would link the individual figure panels a bit more closely to their text. At times I felt like I was reading two papers in parallel, especially as the figure panels are not well enough labeled as to not require reading the captions*.

We thank the reviewer for this suggestion. We revised the text in several places to improve clarity.

*4) In the sliding window ROC analysis, the authors find that 28/29 SERT+ neurons are phasically activated*. *What does this mean? Meaning that they had 1 bin above chance across the analysis? How many bins and what p level is this?*

Yes, the significance was measured bin by bin, with P < 0.05 (after Bonferroni correction for multiple comparisons). We clarified this in the Methods.

*5) In the Results, there is a paragraph describing the response of the neurons to unexpected reward and punishment. It comes a bit out of the blue. I was not clear even where these data came from in the context of the task. I think this should be more clearly explained. I'd also like to see the activity directly compared to the expected. This will make more clear whether there is an implication of error signaling here or if it is just that the neurons are firing to the US the same way they are during the trials*.

We thank the reviewer for this suggestion. We revised the text to better motivate this analysis and directly compared the responses to expected versus unexpected reward (subsection “Serotonergic neurons are phasically excited or inhibited by reward-predicting cues or punishments” and new Figure 4). As a population, serotonergic neurons showed a weak tendency to fire more during unexpected rewards compared to expected ones, whereas dopaminergic neurons showed this more strongly.

*6) Related to point #5, I think that unless you find clear evidence that the neurons fire to unexpected reward or punishment differently from how they fire to expected, then you should not emphasize this point in the seventh paragraph of the Discussion. There you start by emphasizing the response. To me this carries the connotation of error signals. This has not been shown and Doya et al. have provided strong data against this. And I think the results here are largely in agreement. Assuming this is not the intent here, this might be modified. (If it is the intent, then I think much more must be done to show that these are error signals*.*)*

We agree that it was a bit misleading to compare serotonergic responses to dopaminergic responses in the way we did. We modified the Discussion accordingly.

Reviewer #2:

*In this very interesting study, Cohen et al. provide what is the most comprehensive characterization to date of the activity of optogenetically-identified dorsal raphe serotonergic cells in a Pavlovian paradigm involving predictions of rewards and punishments. The results are quite complex, with differential modulations over multiple timescales, and not complete. However, they represent an important step forward*.

*Critical questions*:

*1) We need to know a bit more about the behaviour of the animals: do they orient in any way to all three types of cues, is this response related to the 5-HT activity? Can they avoid sampling the odour in the aversive blocks? Do they show other changed behavior? Does the licking response show anything like the topography seen in*
Figure 3—figure supplement 2*? This is important given evidence from the likes of Barry Jacobs about activity-related influences on 5-HT firing*.

We thank the reviewer for this comment. Because mice were head-fixed (we apologize if this was not clear before; we added text to make this clearer), we did not measure orienting behavior. They could not avoid sampling the odor during aversive blocks. We added a new analysis in which we compared the licking to the firing rates in Figure 3—figure supplement 2. We found that the lick rates were relatively stable within blocks, in contrast to the build-up and build-down of firing rates.

*2) I was confused by the status of the neutral trials. In the Methods, it sounds as if neutral stimuli were interspersed with probability 0.1 in each block (also by reading of*
Figure 1*), in which case I'd like to see what happens on those trials. But later in the paper, it sounds as if there were whole blocks with neutral trials too. In the latter case, if these blocks were relatively unusual, then I think it hard to make any argument from them about salience versus value coding, since they could be salient by virtue of being rare*.

We apologize for the confusion. We described different tasks in different parts of the manuscript to make different points. We began with the task in Figure 1, in which reward and punishment blocks each contained 0.1 probability of a neutral trial.

Later, we wanted to demonstrate that tonic serotonergic firing is value-dependent. Thus, we modified the task to include blocks of neutral trials (with one-third probability). We modified the text to make this clearer, and added a new Figure 6—figure supplement 1.

*3) I was disappointed at how little analysis there was of the data in*
Figure 6
*when there was an additional level of reward. It is important to report what happens to other aspects of the activity of the 5-HT neurons, for instance, how are the prominent value-related CS responses affected? How does the monotonic relationship in*
Figure 6
*relate to baseline activation by reward/punishment? The logic of these analyses is the same as the logic of the authors having changed reward values in the first place*.

We thank the reviewer for these questions. We added analyses to Figure 6—figure supplement 2.

*4) Schwiemer et al. reported different classes of physiological activity for their 5-HT neurons, and that the response to punishment was related (e.g. 'clock-like'). Was there any evidence of that here, or could that have arisen from the anaesthetic*?

We thank the reviewer for this question. We added to the text to the subsection “Unidentified neuron responses” to address this.

*5) Was there any modulation of the baseline or CS activity as a function of the length of the ITI (it would be good to show an interquartile split)? This directly bears on the nature of prediction revealed by these neurons*.

We tested this and found no significant difference between tonic modulation or CS activity as a function of ITI duration, and report this lack of correlation in the subsection “Tonic firing modulation by long-term values”.

*6) The Introduction is a bit woeful compared with the care in the rest of the paper. Tops et al. isn't correctly represented, the stimulation work on patience and preference and optogenetic activation of 5-HT neurons should be better recognized; likewise the heterogeneity revealed by cFOS imaging of the raphe (from Lowry, Maier and others). Also, there is not really a competition between tonic and phasic aspects of dopamine or serotonin signaling. These are different facets of the signal that could even be read by different downstream mechanisms. This comes across more reasonably in the Discussion*.

We thank the reviewer for this comment. We revised much of the Introduction accordingly.

*7) The structure of the methods in the Results was a bit puzzling, not only the issue of neutral trials/blocks mentioned above, but also not thoroughly discussing the methods for the DA part of the study*.

We apologize for the confusion and revised the text to make this clearer.

Reviewer #3:

This study from Cohen et al. investigates the phasic and tonic responses of optogenetically identified serotonergic neurons in the raphe nucleus to rewarding and aversive stimuli. They find that, unlike dopaminergic neurons, serotonergic neurons exhibit long-lasting tonic signals that track the value of the entire block. Serotonergic neurons also respond phasically to rewarding and punishing stimuli, with stronger phasic responses to the CS on rewarded blocks and to the US on punished blocks. Overall the experiments are well designed and executed, and the analyses are sound. However, several points require further explanation or clarification as detailed below:

*1) The authors should clarify the methodology used to identify serotonergic neurons; namely the* “*light-evoked energy*” *analysis in*
Figure 2
*should be explained more thoroughly, as it is not a standard method*.

We clarified this approach in the subsection “Identifying serotonergic neurons”.

*2) The authors should include more discussion about why the neural responses to punishment occur consistently later than those to reward. Why do they not respond phasically to the punishment-predicting cues but only the reward-predicting cues*?

We thank the reviewer for this comment. We added a further analysis to the Discussion.

*3)*
Figure 6
*requires more explanation. The plot in panel (a) does not very convincingly convey the conclusions the authors state, and it is confusing that in panel (b) a larger reward is correlated with decreased activity in the serotonergic cells*.

We thank the reviewer for this important point. We agree that the more appropriate statement is that we could not find evidence for salience coding with tonic firing rates (panel (a)). Thus, we modified the text. In panel (b), we show tonic activity for both positive-coding and negative-coding serotonergic neurons (cf. Figure 3), and found, remarkably, that both types showed monotonic firing rates as a function of reward size (albeit some with positive slope, some with negative slope).

*4) Are there differences in the anatomical location within the raphe of cells that respond preferentially to reward vs. punishment*?

This is an important question, but one that, unfortunately, we cannot address with our recording technique. Juxtacellular recordings, such as those from Ungless’s group, are better-suited to address this question. We did, however, find neurons in close proximity that displayed very different responses.

*5) The authors should discuss their results in light of recently published work employing optogenetic stimulation of serotonergic neurons in different reward-related behavioral paradigms (e.g.*
[66]
*Neuron;*
[74]
*Cell Rep.)*.

We thank the reviewer for this comment. We added to the text accordingly.